# Myocardial extracellular volume quantification by computed tomography predicts outcomes in patients with severe aortic stenosis

**Yoav Hammer[1,2◉], Yeela Talmor-Barkan[1,2,3◉], Aryeh Abelow[1,2], Katia Orvin[1,2], Yaron Aviv[1,2], Noam Bar[3], Amos Levi[1,2], Uri Landes[1,2], Gideon Shafir[1,2], Alon Barsheshet[1,2], Hana Vaknin-Assa[1,2], Alexander Sagie[1,2], Ran Kornowski[1,2], Ashraf Hamdan[1,2]***

**1** Department of Cardiology, Rabin Medical Center – Beilinson Hospital, Petach Tikva, Israel, **2** Sackler Faculty of Medicine, Tel Aviv University, Tel Aviv, Israel, **3** Department of Computer Science and Applied Mathematics, Weizmann Institute of Science, Rehovot, Israel

◉ These authors contributed equally to this work.
* hamdashraf@gmail.com

**Data Availability Statement:** All relevant data are within the manuscript and its Supporting information files.

## Abstract

### Background

The extent of myocardial fibrosis in patients with severe aortic stenosis might have an important prognostic value. Non-invasive imaging to quantify myocardial fibrosis by measuring extracellular volume fraction might have an important clinical utility prior to aortic valve intervention.

### Methods

Seventy-five consecutive patients with severe aortic stenosis, and 19 normal subjects were prospectively recruited and underwent pre- and post-contrast computed tomography for estimating myocardial extracellular volume fraction. Serum level of galectin-3 was measured and 2-dimensional echocardiography was performed to characterize the extent of cardiac damage using a recently published aortic stenosis staging classification.

### Results

Extracellular volume fraction was higher in patients with aortic stenosis compared to normal subjects (40.0±11% vs. 21.6±5.6%; respectively, p<0.001). In patients with aortic stenosis, extracellular volume fraction correlated with markers of left ventricular decompensation including New York Heart Association functional class, left atrial volume, staging classification of aortic stenosis and lower left ventricular ejection fraction. Out of 75 patients in the AS group, 49 underwent TAVI, 6 surgical AVR, 2 balloon valvuloplasty, and 18 did not undergo any type of intervention. At 12-months after aortic valve intervention, extracellular volume fraction predicted the combined outcomes of stroke and hospitalization for heart failure with

**Funding:** The author(s) received no specific funding for this work.

**Competing interests:** The authors have declared that no competing interests exist.

an area under the curve of 0.77 (95% confidence interval: 0.65–0.88). A trend for correlation between serum galectin-3 and extracellular volume was noted.

## Conclusion

In patients with severe aortic stenosis undergoing computed tomography before aortic valve intervention, quantification of extracellular volume fraction correlated with functional status and markers of left ventricular decompensation, and predicted the 12-months composite adverse clinical outcomes. Implementation of this novel technique might aid in the risk stratification process before aortic valve interventions.

## Introduction

Myocardial diffuse interstitial fibrosis (DIF) is recognized as a pathological process which characterizes a variety of cardiovascular diseases [1], and is associated with adverse cardiovascular events [2]. DIF involves the interstitial tissue located between myocytes or myocyte bundles, and can be estimated by measuring extracellular volume (ECV) fraction. Histological sample is the most accurate method to estimate the extent of fibrosis [3]. However, this method is invasive and mainly limited by the small sample that can be obtained. Hence, a non-invasive estimation of myocardial DIF by ECV fraction quantification is preferred.

Previous studies have evaluated myocardial fibrosis mainly by magnetic resonance imaging (MRI)-based late gadolinium enhancement (LGE) [4, 5]. Gadolinium-enhanced cardiovascular magnetic resonance is used for differentiation of heart failure related to dilated cardiomyopathy and coronary artery disease. However, DIF is difficult to distinguish using LGE [6]. In recent years, it has been shown that ECV can be derived from pre-contrast computed tomography (CT) and delayed-phase CT with iodinated contrast medium [7–9].

In the context of severe aortic stenosis (AS), the degree of DIF was shown to play an important role in the transition from well-compensated hypertrophy, to overt heart failure and risk of sudden cardiac death [10]. Furthermore, previous MRI studies found the presence of LGE, indicating focal fibrosis, is an independent predictor of mortality in patients with AS that could aid in optimizing the time for surgical aortic valve replacement (AVR) [11–13].

A variety of cytokines, chemokines, and angiogenic factors participate in the regulation and activation of pro-fibrotic processes [14]. Galectin-3 (Gal-3) is a pleiotropic lectin that activates a multiple pro-fibrotic factors, promotes fibroblast proliferation and transformation, and mediates collagen production, with recent evidence suggesting key roles for Gal-3 in fibrogenesis in diverse organ systems, including the myocardium [15].

The implication of ECV fraction using CT in patients with severe AS scheduled for aortic valve interventions was only recently introduced and focused mainly on patients with amyloidosis and patients with low flow low gradient AS [16–18].

Therefore, we aimed to: (1) Compare CT ECV fraction to functional status and extent of imaging parameters of cardiac damage, and to assess its correlation with Gal-3; and (2) evaluate the impact of CT ECV fraction on clinical outcomes in patients with severe AS undergoing CT before aortic valve interventions.

## Materials and methods

### Study cohort

We prospectively recruited 75 patients with symptomatic severe AS who were candidates for AVR (AS group) and 19 subjects without evidence of cardiovascular disease (control group) in the time period between 2016 and 2018. All patients were referred for CT angiography by their caregivers—AS group patients were referred as part of the routine evaluation before AVR in our center, and the control group patients were healthy subjects referred to CT to rule out coronary artery disease. All patients had to give their consent for an additional post contrast scan that was performed immediately after the routine scan was done. While initially recruited 85 patients with symptomatic severe AS, 10 patients were excluded due to pacemakers, implantable cardioverter defibrillators, metallic foreign objects in the proximity of the heart, and surgical aortic valve replacement. All of these may lead to beam-hardening artifacts, which may degrade the inaccuracy in the ECV measurements. Subjects enrolled in the study were at least 50 years of age, in order to avoid radiation exposure in younger patients who have a higher lifetime attributable risk than older individuals receiving the same dose.

Severe aortic stenosis was defined as per contemporary valvular heart disease guidelines (symptoms typical of aortic stenosis, mean aortic gradient > 40 mmhg, valve area < 1.0 $cm^2$) [19]. The decision whether to pursue aortic valve implantation was dependent upon the decision of a dedicated consultation 'heart team' forum that includes a multidisciplinary team of clinical cardiologists, imaging specialists, interventional cardiologists, cardiac surgeons, and geriatricians as required.

Blood samples for measuring hematocrit and serum level of Gal-3 were drawn from all participants immediately prior to CT scan. The study was approved by the Ethics Committee of the Rabin Medical Center, and all patients signed a written informed consent.

### CT data acquisition and reconstruction

All patients underwent CT angiography as a part of a routine evaluation before TAVI using a 256-slice system (Brilliance iCT, Philips Healthcare, Cleveland, Ohio). Pre-contrast CT scans had a tube voltage of 120kV, tube current 337 mAs, and gantry rotation time of 330ms. Acquisition was performed during an inspiratory breath hold, while the electrocardiogram was recorded simultaneously to allow for prospective gating of the data (75 percent of the RR interval). Mean heart rate for the study cohort was 72.1 beats per minute (±10.9).

Besides the pre-contrast scan, an additional post-contrast scan with the same scan parameters was added 7 minutes after contrast infusion, which added a radiation dose of about 1.5 millisiverts. Images were reconstructed using iterative model reconstruction (level 2) with a slice thickness of 2 mm.

In order to avoid contrast agent exposure and potential risk of contrast nephropathy in elderly patients we used intravenous injection of 50–60 ml of nonionic contrast agent (Iopromide 370, Bayer Shering, Berlin, Germany) at a flow rate of 3 ml/s was followed by a 30-ml saline chase bolus (3 ml/s). The 3-dimensional dataset of the contrast-enhanced scan was reconstructed at 10 percent intervals over the cardiac cycle, generating a 4-dimentional CT dataset, which was used to assess the maximal left atrial volume. In addition, left ventricular end-diastolic and end-systolic volumes were obtained to calculate left ventricular ejection fraction.

### ECV fraction calculation

Myocardial and blood pool attenuation values were measured in the pre-contrast and post-contrast CT scan by a reader experienced in cardiovascular imaging, who was blinded to the

**Fig 1. Representative example of extracellular volume (ECV) fraction measurement.** Myocardial and blood pool attenuation values were measured at the mid segment of the septum and at the descending aorta; respectively, in the pre-contrast (A) and 7 minutes post-contrast (C) CT scan. The contrast scan (B) was used to localize the region of interest, which than could be replicated to the pre- and post-contrast scan. Attenuation values were than used to calculate ECV fraction.

clinical data. Region of interest (ROI) was drawn on myocardial septum with the greatest area and within the aorta. In cases where ECV measurement in the myocardial septum was felt to be possibly inaccurate (e.g thin septum, beam hardening or streak artifacts), ROI was drawn at lateral wall. If there was uncertainty about which one has the most accurate ECV, an average of these two was calculated. The ROI's were first drawn on the CT scan obtained during contrast injection, and then copied to the pre- and post-contrast CT scans (Fig 1). Care was taken to avoid ECV measurement in myocardial segments with previous myocardial infarction.

Mean attenuation in Hounsfield units (HU) of the myocardium and blood was then recorded and myocardial ECV fraction calculated using the formula:

$$ECV = (1 - hematocrit\ )X\left(\frac{\Delta HU_{tissue}}{\Delta HU_b}\right)$$

where $\Delta HU_{tissue}$ is the change in attenuation of the myocardium, and $\Delta HU_b$ is the change in attenuation of the blood. The change in attenuation was determined by the following formula:

$$\Delta HU = HU_{post} - HU_{pre}$$

where $HU_{post}$ and $HU_{pre}$ are attenuation after and before contrast agent infusion, respectively.

## Serum galectin-3 measurement

Venous blood samples were taken from patients just before the CT scan. Gal-3 level was measured in serum samples. All samples were allowed to clot for 30 minutes at room temperature before centrifugation for 15 min at 1000 x g, then the collected supernatant was stored at -80˚C until biomarker measurement. Gal-3 level was measured by using a high sensitivity, quantitative sandwich enzyme immunoassay from R&D Systems Inc. (Minneapolis, MN, USA), according to manufacturer's instructions.

## AS staging classification

All patients underwent a 2-dimentional echocardiography, and were then stratified into five stages according to the extent of cardiac damage, as previously described by Généréux et al. [20]: Stage 0: no cardiac damage detected; Stage 1: left ventricular (LV) damage as defined by presence of LV hypertrophy (LV mass index >95 g/m$^2$ for women, >115 g/m$^2$ for men), severe LV diastolic dysfunction (E/e' >14), or LV systolic dysfunction (LV ejection fraction <50%); Stage 2: left atrial (LA) or mitral valve damage or dysfunction as defined by the

presence of an enlarged left atrium ($>34$ mL/m$^2$), presence of atrial fibrillation, or presence of moderate or severe mitral regurgitation; Stage 3: pulmonary artery vasculature, or tricuspid valve damage or dysfunction; defined by the presence of systolic pulmonary hypertension (systolic pulmonary arterial pressure $>60$ mmHg) or presence of moderate or severe tricuspid regurgitation; Stage 4: right ventricular (RV) damage as defined by the presence of moderate or severe RV dysfunction. Due to inadequate echocardiography image quality, staging classification of AS was performed in 66 out 75 patients.

## Statistical analysis

Continuous variables are expressed as mean ± standard deviation or number and percentage, and were compared using Mann-Whitney test. Categorical variables were compared using chi-square or Fisher's exact test as needed. Pearson correlation was used to assess statistical correlation between ECV fraction and clinical, echocardiographic parameters, and serum level of Gal-3. Spearman correlation was used to assess correlation between ECV fraction and NYHA functional class. Logistic regression models were used to evaluate clinical outcomes. The ability of ECV to predict the combined clinical outcomes of stroke, hospitalization for heart failure, and all-cause mortality at 12 months after aortic valve intervention was evaluated using the receiver-operator-characteristic curve. Clinical outcomes were calculated only for patients who underwent an aortic valve intervention. Inter-observer variation analysis was performed using Bland-Altman methods and calculating the correlation coefficient. Statistical analysis was performed using the Python statistical software package, version 3.5. A p-value of $<0.05$ was considered statistically significant.

## Results and discussion

Baseline characteristics of all participants are summarized in Table 1. Patients in the AS group were older and had more comorbidities compared with the controls. Of the 75 patients in the

**Table 1. Baseline characteristics.**

| Characteristics | Aortic stenosis (n = 75) | Normal subjects (n = 19) | P value |
|---|---|---|---|
| Age, years | 80.6 ± 6.8 | 57.2 ± 5.6 | <0.001 |
| Gender, female | 36 (48.0) | 12 (63.2) | 0.31 |
| BMI (kg/m$^2$) | 27.5 ± 5.2 | 27.6 ± 5.3 | 0.38 |
| Hypertension | 62 (82.7) | 5 (26.3) | <0.001 |
| Diabetes mellitus | 32 (42.7) | 3 (15.8) | 0.035 |
| Hyperlipidemia | 62 (80) | 10 (52) | 0.02 |
| Current smoking | 0 (0.0) | 1 (5.3) | 0.2 |
| Previous smoker | 10 (13.3) | 5 (26.3) | 0.18 |
| Coronary artery disease | 35 (46.7) | 2 (10.5) | 0.003 |
| Previous MI | 6 (8.0) | 0 (0.0) | 0.3 |
| Previous CABG | 9 (12.0) | 0 (0.0) | 0.2 |
| Previous PCI | 25 (33.3) | 0 (0.0) | 0.002 |
| Atrial fibrillation | 12 (16.0) | 1 (5.3) | 0.4 |
| COPD | 6 (8.0) | 0 (0.0) | 0.3 |
| eGFR (ml/min) | 68.3 ± 20.3 | 97.3 ± 26.4 | <0.001 |
| NYHA FC | 2.3 ± 0.6 | 1.0 ± 0.0 | <0.001 |

Values are presented as n (%) or mean ± SD.

BMI: body mass index; CABG: coronary artery bypass graft; COPD: chronic obstructive pulmonary disease; eGFR: estimated glomerular filtration rate; MI: myocardial infarction; NYHA FC: New York Heart Association functional class; PCI: percutaneous coronary intervention.

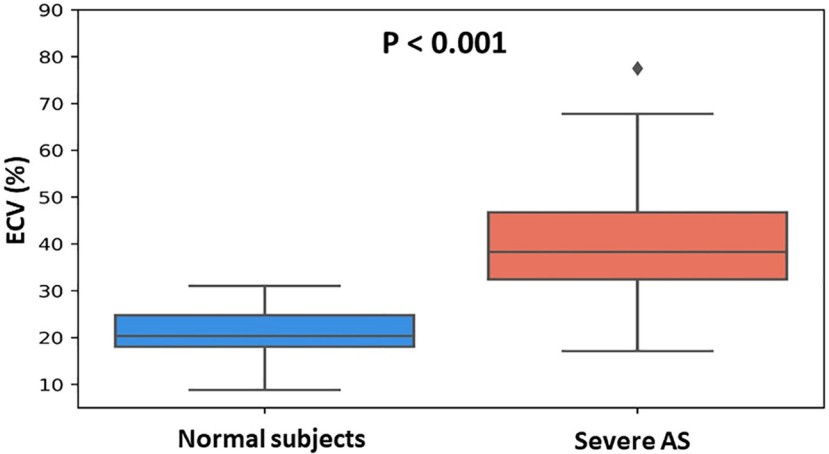

**Fig 2. ECV fraction of patients with severe aortic stenosis and normal subjects.** Extracellular volume (ECV) fraction was significantly higher in patients with severe aortic stenosis than in normal subjects.

AS group, 49 underwent TAVI, six surgical AVR, two balloon valvuloplasty, and 18 did not undergo any type of intervention during the time period of data collection: nine patients were ultimately denied intervention after a heart team discussion, five refused to undergo any intervention, two died before intervention, and two were lost to follow-up.

Mean ECV fraction in patients with AS was significantly higher than in the control group (40.0±11.0% vs. 21.6±5.6%, respectively, p <0.001) (Fig 2). A histogram of ECV distribution in the AS group is presented in Fig 3. To note, ECV fraction did not differ between patients who underwent and those who did not undergo aortic valve intervention (38.7±6.9% vs. 40.0 ±11.0%, respectively, p = 0.2). Multivariate analysis using linear regression showed that ECV fraction was significantly higher in patients with AS compared to the control group, independent from age, gender, BMI, diabetes mellitus, and hypertension.

ECV fraction correlated with New York Heart Association (NYHA) functional class (FC) at baseline (r = 0.30, p = 0.01), as well as with CT- and echocardiographic-based markers of LV

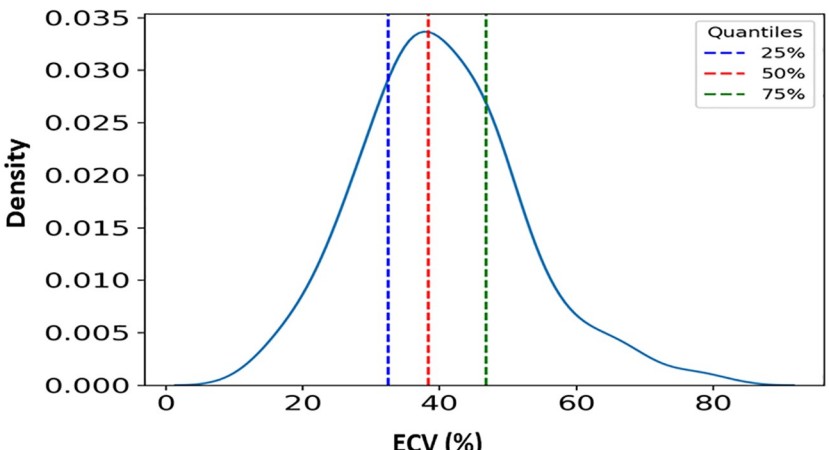

**Fig 3. ECV fraction distribution in patients with symptomatic severe AS.** Mean Extracellular volume (ECV) fraction was 40% in patients with severe aortic stenosis, and 50% of patients (Q1-Q3) had an ECV fraction between 30.5% and 47.8%.

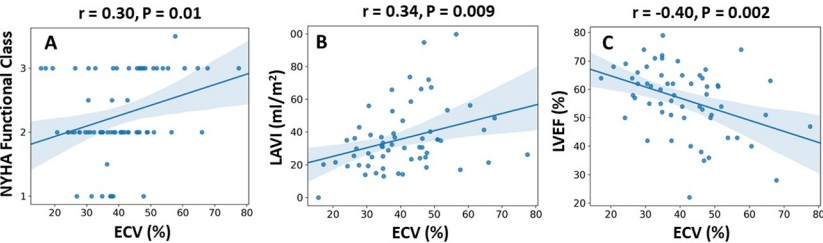

**Fig 4. The relation between ECV fraction and functional/imaging parameters.** Extracellular volume (ECV) fraction correlated positively with New York Heart Association (NYHA) functional class (A) and left atrial volume index (LAVI) (B), and negatively with left ventricular ejection fraction (LVEF) (C).

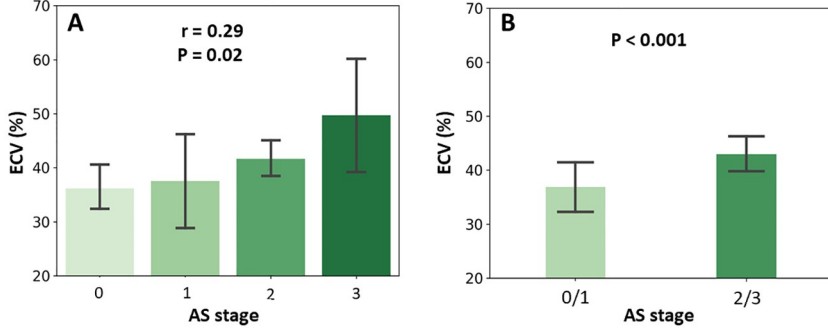

**Fig 5. The relation between ECV fraction and staging classification of AS, based on the extent of cardiac damage.** (A) In a Pearson correlation model, extracellular volume (ECV) fraction correlated significantly and progressively with aortic stenosis (AS) stage. To note, eight patients were at stage 0, nine patients at stage 1, 41 patients at stage 2, eight patients at stage 3, and none at stage 4. (B) In a logistic regression model, patients in stages 2 or 3 had a significantly higher ECV fraction compared to patients at stages 0 or 1.

decompensation including left atrial volume (r = 0.34, p = 0.009), lower LV ejection fraction (r = -0.40, p <0.001) (Fig 4), and increasing extent of cardiac damage (r = 0.29, p = 0.002) (Fig 5).

At 12-months after aortic valve intervention, ECV fraction correlated positively and significantly with NYHA FC (r = 0.6, p<0.001) (Fig 6B), and was significantly higher among patients hospitalized for heart failure (Fig 6A).

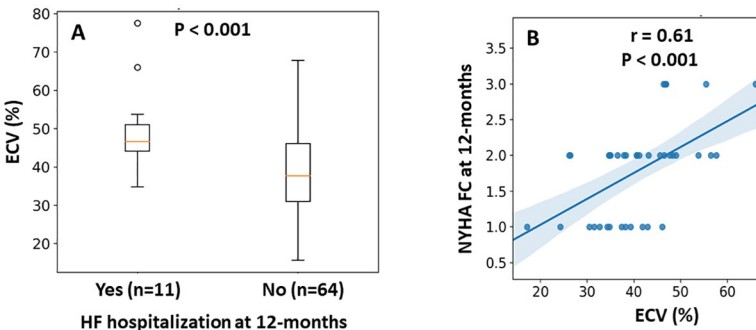

**Fig 6. The relation between ECV fraction and clinical outcomes.** (A) In a logistic regression model, extracellular volume (ECV) fraction in patients hospitalized for heart failure (HF) at 12-months was significantly higher than in those who were not. (B) Increased ECV fraction was associated with worse functional class at 12 months (r = 0.6, p<0.001). To note, there were no patients with NYHA FC 4 in our cohort. NYHA = New York Heart Association; FC = Functional class.

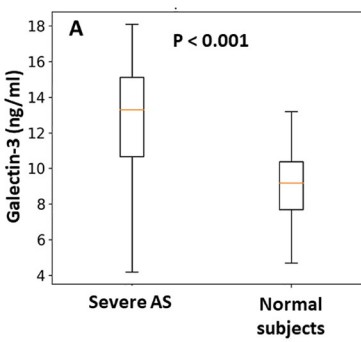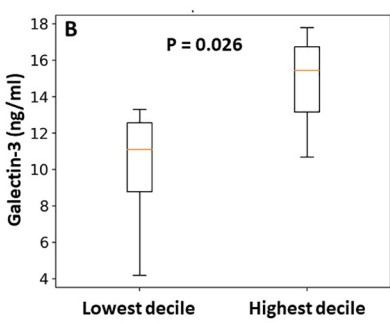

**Fig 7. Galectin-3 level in patients with severe AS.** (A) Galectin-3 level (ng/ml) was significantly higher in patients with severe aortic stenosis (AS) than in normal subjects. (B) Galectin-3 level was significantly higher in the highest decile of extracellular volume (ECV) fraction (ECV >57%) versus the lowest decile (ECV <26%).

Gal-3 level in patients with AS was significantly higher than in normal subjects (12.9±3.1 ng/ml vs. 9.1±2.1 ng/ml, respectively, p <0.001) (Fig 7A). Although ECV fraction had a borderline correlation with serum level of Gal-3 (r = 0.22, p = 0.07), patients in the highest decile of ECV fraction (ECV >57%) had significantly higher levels of Gal-3 compared to patients in the lowest decile (ECV <26%), (15.7 ng/ml vs. 11.3 ng/ml, respectively, p = 0.02) (Fig 7B). In addition, Gal-3 trended towards a positive correlation with echocardiographic based AS stages (r = 0.23, p = 0.08).

At 12-months, 14 adverse cardiac events occurred in 57 patients who underwent an aortic valve intervention: hospitalization for heart failure in eleven patients (19%) and stroke in three patients (5%). There were no deaths. ECV fraction predicted the 12-months combined clinical endpoints of stroke and hospitalization for heart failure, with an area under the curve of 0.77 (95% confidence interval: 0.65–0.88) (Fig 8). Sensitivity/specificity plots yielded an ECV fraction of 40.8% as the optimal cutoff points for prediction of clinical outcomes, with a sensitivity of 91% and specificity of 64%. Patients in the highest quartile of ECV fraction (ECV >47.8%) had the highest likelihood for the combined clinical outcomes of stroke and hospitalization for heart failure at 12-months, compared to patients with an ECV in the lowest quartile (ECV <30.5%), (31.6% vs. 10.1%, p = 0.02).

In paired sample comparison of 20 randomly selected patients with AS, interobserver measurements of ECV showed an interclass correlation coefficient of 0.93, p <0.001; the paired difference was 0.80±2.7 (95% confidence interval: -4.6 to 6.2).

## Discussion

CT quantification of extracellular space, measured as ECV fraction, was found to correlate with functional status, biochemical and echocardiographic parameters of myocardial damage, and predicted 12-months outcomes of stroke and hospitalization for heart failure among candidate patients with severe AS undergoing CT before aortic valve interventions. ECV fraction measuring requires the addition of a post contrast scan that follows a CT angiography, which is routinely performed in all TAVI candidates.

Although TAVI has emerged as a promising alternative to surgery in patients with severe AS and contraindications to surgery [21], TAVI candidates are typically older and have multiple comorbidities, and thus, serious concerns should be raised about the potential usefulness versus futility of the procedure. Increased myocardial fibrosis may be associated with less favorable TAVI related outcomes, since existing myocardial damage might not fully enable

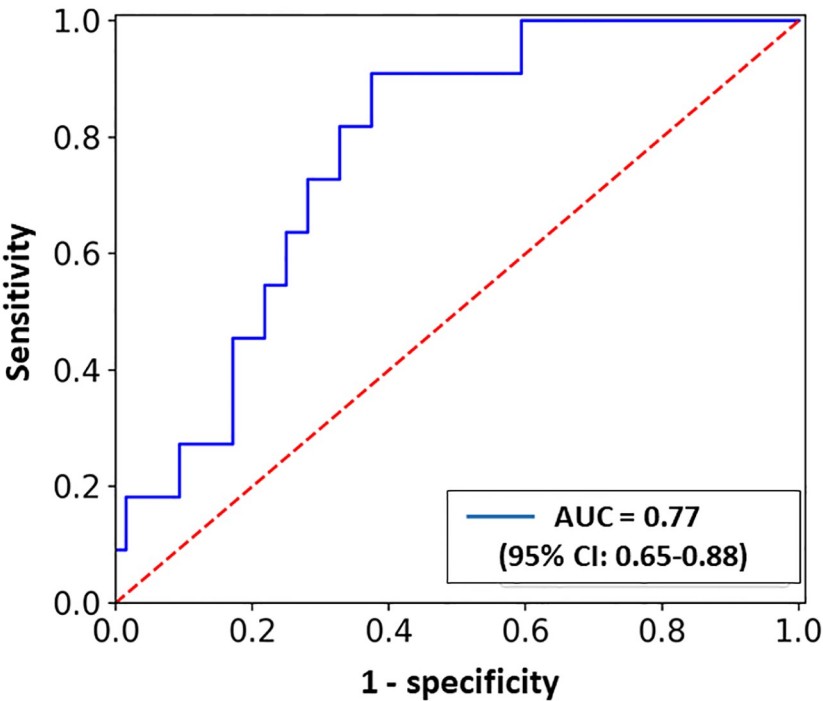

**Fig 8. The ability of ECV fraction to predict clinical outcomes.** Receiver operating curve was used to examine the ability of extracellular volume (ECV) fraction to predict the combined clinical outcomes of stroke and hospitalization for heart failure at 12-months after aortic valve intervention. ECV fraction had a favorable curve with an area under the curve (AUC) of 0.77, 95% confidence interval (CI): 0.65–0.88.

improvement of functional status after the procedure, as previously shown using cardiac MRI [13]. Therefore, ECV fraction measurement in AS patients before surgical or transcatheter aortic valve replacement may help in the stratification process, and may identify those patients with a high degree of DIF who will probably experience less benefit from aortic valve intervention.

Only few studies have addressed the prognostic value of myocardial fibrosis assessment in patients before aortic valve intervention, and most of them used cardiac MRI for that purpose [5, 12, 22]. The literature regarding the use of cardiac CT for quantification of ECV fraction in patients with AS is not robust and had only been recently presented [16–18]. To the best of our knowledge, our study is one of very few studies evaluating myocardial ECV fraction in patients with severe AS, and showing correlation between increased baseline ECV fraction and clinical and echocardiographic parameters. In addition, ECV predicted 12-months combined adverse clinical outcomes of stroke and hospitalization for heart failure among AS patients following aortic valve interventions (Fig 8). Our study also demonstrated that ECV correlated with the recently described staging classification of AS based on the extent of cardiac damage as evaluated by 2-dimentional echocardiography [20], highlighting the importance of ECV in the evaluation of myocardial mutilation among patients with severe AS. Current guidelines recommend risk stratification of patients with AS using the integration of different factors, including the severity of AS, presence of AS-related symptoms, and presence of other risk factors such as STS score, frailty, or compromise of other major organ systems (e.g., kidney disease, lung disease) [19]. However, no recommendation exists on how to incorporate the extent of ECV in clinical decision making related to AS. Given the strong association demonstrated

in our study between ECV fraction and worse clinical outcomes in patients with severe AS, integrating the ECV fraction in the clinical assessment and/or decision-making process before aortic valve interventions might be useful. The demonstrated relation between stroke and diffuse myocardial fibrosis is not clear at this moment, however it might be related to the fact that of the 3 patients who had a stroke, 2 had atrial fibrillation. The existence of atrial fibrillation in those patients might suggest a possible higher degree of myocardial fibrosis.

ECV fraction also correlated with Gal-3 measurements, a validated serum biomarker for fibrosis [15]. Patients in the highest deciles of ECV fraction had a significantly higher serum Gal-3 level than patients in the lowest deciles. The combination of two different methods for assessing and quantifying cardiac damage (ECV and Gal-3) suggests that ECV fraction could be an important factor that might be helpful in the evaluation of patients with symptomatic severe AS prior to intervention.

## Limitations

Our study has the following limitations: First, ECV could be evaluated in only 75 out of 85 AS patients initially enrolled, due to inadequate CT image quality. Second, beam-hardening artifacts originating from aortic valve calcification might affect ECV measurement, and thus special attention was made for determining the exact location of measurement. Third, an important assumption of ECV measurement using CT is that the extracellular space within the tissue consists of a single compartment. In reality, the extracellular space includes an intravascular component in addition to the extracellular space. Where the vascular volume is large, this component may introduce error into ECV measurement. Forth, given a relatively large variability of ECV fraction in our cohort, with some patients having ECV values greater than 60, we did not perform any imaging modality to exclude the coexistence of cardiac amyloidosis. Fifth, while Gal-3 was the main biomarker studied in our study, we did not measure high sensitive troponin and Brain natriuretic peptide (BNP). Sixth, the control group consisted of young healthy patients, and was obviously not matched to the study group in terms of their comorbidities. Last, although measuring ECV fraction only in the septum is a validated and common method, it might not represent the entire myocardium.

## Conclusions

CT estimation of ECV fraction in patients with severe AS correlated with functional status and parameters of myocardial damage, and predicted the combined endpoints of stroke and hospitalization for heart failure. Implementation of this novel technique might aid in the risk stratification process of AS patients before aortic valve intervention.

## Supporting information

**S1 File. Study dataset.**
(XLSX)

## Author Contributions

**Conceptualization:** Yoav Hammer, Yeela Talmor-Barkan, Aryeh Abelow, Yaron Aviv, Noam Bar, Amos Levi, Uri Landes, Gideon Shafir, Alon Barsheshet, Hana Vaknin-Assa, Alexander Sagie, Ran Kornowski, Ashraf Hamdan.

**Data curation:** Yoav Hammer, Yeela Talmor-Barkan, Aryeh Abelow, Katia Orvin, Yaron Aviv, Noam Bar, Amos Levi, Uri Landes, Gideon Shafir, Hana Vaknin-Assa, Ashraf Hamdan.

**Formal analysis:** Yoav Hammer, Yeela Talmor-Barkan, Aryeh Abelow, Yaron Aviv, Noam Bar, Amos Levi, Uri Landes, Gideon Shafir, Alon Barsheshet, Ashraf Hamdan.

**Investigation:** Yoav Hammer, Yeela Talmor-Barkan, Katia Orvin, Gideon Shafir, Alexander Sagie, Ran Kornowski.

**Methodology:** Yoav Hammer, Yeela Talmor-Barkan, Katia Orvin, Yaron Aviv, Noam Bar, Amos Levi, Uri Landes, Gideon Shafir, Alon Barsheshet, Hana Vaknin-Assa, Alexander Sagie, Ran Kornowski, Ashraf Hamdan.

**Project administration:** Yoav Hammer, Alon Barsheshet, Ashraf Hamdan.

**Software:** Yoav Hammer, Aryeh Abelow.

**Supervision:** Katia Orvin, Alon Barsheshet, Hana Vaknin-Assa, Alexander Sagie, Ran Kornowski, Ashraf Hamdan.

**Validation:** Hana Vaknin-Assa, Ran Kornowski.

**Writing – original draft:** Yoav Hammer, Yeela Talmor-Barkan.

**Writing – review & editing:** Aryeh Abelow, Katia Orvin, Yaron Aviv, Noam Bar, Amos Levi, Uri Landes, Gideon Shafir, Alon Barsheshet, Hana Vaknin-Assa, Alexander Sagie, Ran Kornowski, Ashraf Hamdan.

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
