## [Decision Letter · Decision Letter 0]

11 Jan 2021

PONE-D-20-34373

Myocardial extracellular volume quantification by computed tomography predicts outcomes in patients with severe aortic stenosis

PLOS ONE

Dear Dr. Hamdan,

Thank you for submitting your manuscript to PLOS ONE. After careful consideration, we feel that it has merit but does not fully meet PLOS ONE’s publication criteria as it currently stands. Therefore, we invite you to submit a revised version of the manuscript that addresses the points raised during the review process.

We look forward to receiving your revised manuscript.

Kind regards,

Ify Mordi

Academic Editor

PLOS ONE

Journal Requirements:

2.) PLOS requires an ORCID iD for the corresponding author in Editorial Manager on papers submitted after December 6th, 2016. Please ensure that you have an ORCID iD and that it is validated in Editorial Manager. To do this, go to ‘Update my Information’ (in the upper left-hand corner of the main menu), and click on the Fetch/Validate link next to the ORCID field. This will take you to the ORCID site and allow you to create a new iD or authenticate a pre-existing iD in Editorial Manager. Please see the following video for instructions on linking an ORCID iD to your Editorial Manager account: https://www.youtube.com/watch?v=_xcclfuvtxQ

3.) We noticed you have some minor occurrence of overlapping text with the following previous publication(s), which needs to be addressed:

https://pubs.rsna.org/doi/full/10.1148/radiol.13130130

https://academic.oup.com/eurheartj/article/38/45/3351/4002776

In your revision ensure you cite all your sources (including your own works), and quote or rephrase any duplicated text outside the methods section. Further consideration is dependent on these concerns being addressed.

Reviewers' comments:

Reviewer's Responses to Questions

**Comments to the Author**

1. Is the manuscript technically sound, and do the data support the conclusions?

Reviewer #1: Partly

Reviewer #2: Partly

Reviewer #3: Yes

2. Has the statistical analysis been performed appropriately and rigorously? 

Reviewer #1: Yes

Reviewer #2: No

Reviewer #3: Yes

3. Have the authors made all data underlying the findings in their manuscript fully available?

Reviewer #1: Yes

Reviewer #2: No

Reviewer #3: Yes

4. Is the manuscript presented in an intelligible fashion and written in standard English?

Reviewer #1: Yes

Reviewer #2: Yes

Reviewer #3: Yes

5. Review Comments to the Author

Reviewer #1: In this study, ECV was measured using CT in 75 patients with severe AS. ECV was higher in patients with AS compared to normal subjects. ECV correlated with LV dysfunction, NYHA class, LA volume, and AS stage. ECV predicted the combined outcomes of all-cause mortality, stroke and hospitalization for heart failure with an area under the curve of 0.77. A trend for correlation between serum galectin-3 and ECV was noted. Investigation of relationship between Gal-3 and ECV is novel.

This is a clinically important investigation. However, due to following reasons, this reviewer has a major concern in the accuracy of ECV in this study. This reviewer suggests to reanalyze all the data by using global ECV, add analysis of late enhancement and perform a multivariate analysis to find independent predictors of adverse event. Unfortunately, a couple of similar investigations were published elsewhere in the previous months.

1. ECV could be evaluated in only 75 out of 85 AS patients initially enrolled, due to inadequate CT image quality. Methods and results of image quality assessment should be explained.

2. Both pre-contrast and delayed-phase CT appears to be obtained with a standard prospectively-ECG-triggered scan. As has been demonstrated in ref#22, delayed-phase CT may suffer from severe artifact caused by beam-hardening, motion, and partial scan, which may degrade the accuracy of ECV measurement.

3. ECV was measured placing a ROI in the septal wall only. Global ECV or at least averaging of ECV of multiple locations, should more reliably represents the overall extent of LV fibrosis.

4. In AS group, previous MI and revascularizations were common. There is a risk that ECV was measured in infarcted tissue which may not adequately represent the overall fibrosis of the patient. The result of presence and extent of late enhancement should be reported, and discussed in relation to ECV.

5. ECV value in patients with AS varied from less than 20% to larger than 70%. Such high ECV indicates that ECV in the myocardium is almost the same as ECV of the blood pool. ECV of 70-80% should be artifactual. ECV of 60-70% is probably due to artifact or presence of infarction.

6. Presence of infarction that shows high ECV may be the main source of higher rate of adverse events.

7. Wide range of ECV values in normal subject in this study raises suspicion in its accuracy. ECV values in normal subjects (about 26%) has been well established in both CT and MRI as seen in ref#9.

Minor points

Line 33; “newly-defined” is misleading because it gives an impression that the authors suggested a new classification in this manuscript.

Line 78; This sentence is not true anymore.

J Am Coll Cardiol Cardiovasc Imaging. Oct 28, 2020. Epublished DOI: 10.1016/j.jcmg.2020.07.045

J Am Coll Cardiol Cardiovasc Imaging. 2020 Aug, 13 (10) 2177–2189

Line 81; Long-term clinical outcome? With only 12 months of follow up?

Line 101; Scan protocol and reconstruction of pre-contrast and delayed-phase CT should be described in more details. Conversely, CT protocol for coronary CTA can be shortened for this manuscript.

Line 103; Acquisition at 75%RR can suffer from severe motion artifact in case of high heart rate. Please report heart rate during acquisition.

Line 109; Is 50-60ml sufficient for evaluation of delayed enhancement and ECV?

Line 126; Representing the patient’s ECV with a ROI drawn on septum only is probably not adequate.

Line 207; Please clarify what the box and whiskers represent.

Line 296; This is not the first study.

Line 301; “newly described” is misleading.

Reviewer #2: In this manuscript, the authors determined myocardial extracellular volume (ECV) fraction by analyzing pre-contrast and 7 minutes post-contrast CT in 75 patients with aortic stenosis and 16 normal subjects. In the discussion, they stated that 10 of 85 AS patients were evaluated due to inadequate CT image quality. Of the 75 patients in the AS group with successful CT, 49 underwent TAVI, 6 surgical AVR and 2 balloon valvuloplasty, indicating that aortic intervention was performed in 57 patients. Staging of cardiac damage due to AS was successfully performed by echocardiography in 66 of 75 patients.

The major findings in this study were; (1) ECV fraction was significantly higher in patients with AS when compared to control subjects (ECV 40.0±11% vs. 21.6±5.6%; p<0.001, age 57.2±5.6 years vs. 80.6±6.8 years; p<0.001, respectively), (2) ECV fraction can predict combined endpoints of stroke and hospitalization due to heart failure at 12 months with area under ROC of 0.77, (3) The serum galectin-3 and ECV fraction exhibited no significant correlation (r=0.22, p=0.07)

As the authors explained in the discussion, quantification of myocardial ECV fraction by pre- and post-contrast enhanced CT allows for noninvasive assessment of diffuse myocardial fibrosis. High AUC value of 0.77 for the prediction of stroke and hospitalization for heart failure at 12 months demonstrated by using CT ECV fraction is impressive and may have substantial impact for patient care.

Major concerns for this paper are as follows,

1. The aortic interventions were performed in 57 of 75 patients. The prognostic value of CT ECV should be demonstrated by showing Kaplan Meier event-free survival curves in 57 patients who had aortic valve interventions.

2. The staging of AS was successfully performed by echocardiography in only 66 of 75 patients. How were the severity of AS and the indication of aortic valve intervention determined in the patients who enrolled the study?

3. Superiority of CT over CMR for the assessment of myocardial ECV fraction, particularly higher spatial resolution, was too much emphasized in the second paragraph of the introduction. In the current study, the ROI was placed in the interventricular septum, indicating that high spatial resolution and SNR improvement by iterative reconstruction are not relevant. Bean gardening artifacts and motion artifacts are major concerns for CT quantification of myocardial ECV fraction.

4. The authors emphasized that the implication of ECV fraction using CT has never been studied in patients with severe AS. Please revise the introduction and discussion by citing the following publications.

Oda S. Quantification of Myocardial Extracellular Volume With Planning Computed Tomography for Transcatheter Aortic Valve Replacement to Identify Occult Cardiac Amyloidosis in Patients With Severe Aortic Stenosis. Cir Cardiocasc Imaging 2020;13:e010358

Tamarappoo B, et al. Prognostic Value of Computed Tomography-Derived Extracellular Volume in TAVR Patients With Low-Flow Low-Gradient Aortic Stenosis. JACC Cardiovascular Imaging 2020 Oct 28

Scully PR, et al. Identifying Cardiac Amyloid in Aortic Stenosis: ECV Quantification by CT in TAVR Patients. JACC Cardiovascular Imaging 2020 Oct 13.

Specific comments for this paper.

5. Abstract. Similar sentences are repeated for the result and conclusion in the abstract. The presentation of the abstract should be more organized.

6. Introduction. Second paragraph. “Another MRI technique used to quantify DIF is T1-mapping, yet, its key limitation is limited spatial resolution”. The authors put too much emphasis on spatial resolution. At a lower concentration of contrast media, contrast discrimination by X-ray CT is not as good as T1 mapping CMR.

7. Method Study cohort. Please clarify that 10 of 85 AS patients were evaluated due to inadequate CT image quality. Was a prospective study, or a retrospective study that analyzed As patients who had sufficient CT image quality?

8. CT data acquisition and reconstruction. “Besides the pre-contrast scan, a unique post-contrast scan with the same scan parameters was added”. Delete unique. Please add a brief explanation as to why post-contrast images were acquired at 7 minutes.

9. Result. Table 1. There was a significant difference in age between AS patients and control subjects (57.2±5.6 years in control subjects vs. 80.6±6.8 years in AS patients). Previous studies demonstrated that ECV was significantly influenced by age. In addition, prevalence of CAD was significantly higher in the AS group (p=0.003). The differences in age and the degree of diffuse atherosclerosis substantially influenced the ECV.

10. In this paragraph, the authors also stated that multivariate analysis showed that ECV fraction was significantly higher in patients with AS compared to the control group, independent from age, gender, BMI, diabetes mellitus, and hypertension. However, the number of patients was limited to adequately confirm the independency of these parameters.

11. Results. Figure 6B. NYHA FC at 12 months is a categorial parameter and a scatter plot is not suited for Figure 6B.

12. Results. Figure 7. Comparison between the highest decile of ECV and lowest ductile of ECV seems to be arbitral.

13. Page 12. “There were no deaths.” Delete all-cause mortality from the combined endpoints to avoid confusion by the readers.

14. Results. Figure 8. ROC curve. Add cut-off value for ECV.

15. Page 14. Discussion. “Patients in the highest deciles of ECV fraction had a significantly higher serum Gal-3 level than patients in the lowest deciles. The combination of two different methods for assessing and quantifying cardiac damage (ECV and Gal-3) enforced our findings”. Comparison between the highest decile of and lowest ductile was quite subjective and this statement had no sufficient rationale.

Reviewer #3: Study by Hammer et al describes CT ECV quantification for assessment of myocardial fibrosis in severe aortic stenosis and its association with outcome in 75 patients and 19 controls.

General comments : 

1. CTecv was calculated in the septum pre and post contrast as perviously described in CMR and CT manuscripts. As it is supposed to reflect diffuse myocardial fibrosis, the basal septum should be representative as it is relatively spared from myocardial infarction. With that in mind, I am very concerned about the distribution of the CTecv with 11 patients having an ECV >50% and a majority an ECV >40%. This amount of interstitial expansion is rather unusual unless there is evidence of myocardial infiltation (e.g. cardiac amyloidosis). The alternative is a methodological error.  

2. The abstract should include structured headings, and the authors should include a heading for the discussion, which is missing. 

3. The authors have chosen Galactin-3 rather than established cardiac biomarkers like BNP or troponin in this paper. Although this is interesting, the lack of BNP and troponin feels like an oversight. 

4. The combined clinical endpoint of all-cause mortality, stroke and hospitalization for heart failure is misleading and not ideal: -- although all cause mortality and HHF are adequate endpoints, the link between diffuse fibrosis and stroke is not clear. -- no deaths occured, only HHF and stroke -- the highest quartile of ECV chosen here was ECV>47.8% which is unusually high, and can only be explained by amyloidosis (see above). 

5. Regarding endpoints, it is unclear how many endpoints occured periprocedural. 

6. It is nice to see that the authors made full use of the CT dataset and analysed LV size and function as well as atrial volumes. 

Specific comments: 

-Abstract: 

-- The authors need to make it clear in the abstract that on 57/75 patients underwent aortic valve intervention (and what intervention TAVR vs SAVR vs valvuloplasty). 

- Introduction: 

-- The CMR section does not mention ECV quantification (just T1 mapping, which is used for ECV calculation) - this is mentioned in the first paragraph without mentioning the associated imaging modalities (CMR and CT). 

-- The technological improvements in CT are for certain, though relevant to ECV quantification is not temporal resolution, but Hounsfield unit stability as well as signal to noise ratio. 

-- When discussing CMR predictors of outcome in AS, please add ECV by CMR (Everett et al JACC 2020), which is probably more relevant than the LGE papers. 

- Methods: 

-- Please specify ethics committee reference and the period of recruitment. -- please specify which contrast agent you administered. 

-- please provide more details for the control cohort. Did this cohort undergo CT for a clinical indication or purely for research? This cohort is neither age nor sex matched, and have significantly less hypertension, CAD and renal impairment than the AS cohort. 

- Results:-- Please restructure this and place the figure legends to the end of the manuscript. -- how many patients had pacemaker or ICDs? These are more likely to affect myocardial HU then AoV Calcification.

- Discussion: 

-- please add a heading and restructure the discussion to follow a logical flow. 

-- Please add reference and discuss  papers on CTecv by Tamarapoo and Scully that address CTecv in TAVI and outcome (both JACC Imaging).

6. PLOS authors have the option to publish the peer review history of their article (what does this mean?). If published, this will include your full peer review and any attached files.

Reviewer #1: No

Reviewer #2: No

Reviewer #3: No

---

## [Author Response · Author response to Decision Letter 0]

22 Jan 2021

Editor

Comment: minor occurrence of overlapping text.

Response: 

1. For "Staging classification of aortic stenosis based on the extent of cardiac damage" by Phillip Genereux et al – As far as we noticed, the overlapping text is the definitions of the different stages of AS. These definitions are clear and cannot be changed. This overlap appears only in the methods section and is backed up with an appropriate reference. Please let us know if there are any overlaps with this paper or if the editor would like us to change it somehow.

2. For "Measurement of Myocardial Extracellular Volume Fraction by Using Equilibrium Contrast-enhanced CT: Validation against Histologic Findings" by Steve Bandula et al – Unfortunately we could not identify the overlap between this paper and our paper. We would highly appreciate your guidance regarding what are the exact parts of our paper which overlapped, so we could address it. 

Reviewer 1:

Thank you for your helpful comments and suggestions. Below please find the responses that were incorporated into the manuscript.

Comment: "ECV could be evaluated in only 75 out of 85 AS patients initially enrolled, due to inadequate CT image quality. Methods and results of image quality assessment should be explained"

Response: We thank the reviewer for this comment. In order to make it clear we rephrased the 1st paragraph of the methods section, and added the following text:

"While initially recruited 85 patients with symptomatic severe AS, 10 patients were excluded due to pacemakers, implantable cardioverter defibrillators, metallic foreign objects in the proximity of the heart, and surgical aortic valve replacement. All of these may lead to beam-hardening artifacts, which may degrade the inaccuracy in the ECV measurements". (Materials and methods section, study cohort subsection, page 5)

Comment: "Both pre-contrast and delayed-phase CT appears to be obtained with a standard prospectively-ECG-triggered scan. As has been demonstrated in ref#22, delayed-phase CT may suffer from severe artifact caused by beam-hardening, motion, and partial scan, which may degrade the accuracy of ECV measurement"

Response: We fully agree with the reviewer, therefore 10 patients with metallic devices were excluded from the study cohort in order to make our measurement accurate and reproducible. 

Comment: "ECV was measured placing a ROI in the septal wall only. Global ECV or at least averaging of ECV of multiple locations, should more reliably represents the overall extent of LV fibrosis"

Response: We thank the reviewer for this comment. Indeed, global ECV calculation is now available with a dedicated software. Measurement in the myocardial septum is an accepted method of calculating ECV. To note, our protocol when drawing ROI was that in cases where ECV measurement in the myocardial septum was felt to be possibly inaccurate (e.g thin septum, beam hardening or streak artifacts), than ROI was drawn at lateral wall. If there was uncertainty about which one has the most accurate ECV, an average of these two was calculated. This sentence was added to the methods section (Materials and methods, page 7, 2nd paragraph).

Moreover, ECV is a diffuse process, and thus ECV measurement in different segments of the left ventricle should be similar. The septum, which is usually the thickest segment of the myocardium, was felt to be the most accurate segment for ECV calculation.

Comment: "In AS group, previous MI and revascularizations were common. There is a risk that ECV was measured in infarcted tissue which may not adequately represent the overall fibrosis of the patient. The result of presence and extent of late enhancement should be reported, and discussed in relation to ECV"

Response: We thank the reviewer for this comment. Only 6 patients of the entire cohort had previous MI. The average ECV of those 6 patients is 37%, which is lower compared with the average ECV (40%) in our study cohort. Moreover, none of these patients had an ECV above 46%. Since their average ECV is considered low, we are not concerned with ECV overestimation in those patients. As mentioned earlier in this letter, if the septum appeared to be scarred, or the septum measurement was felt to be inaccurate, ECV measurement was taken at the lateral wall. To note, we carefully avoid ECV measurement in LV segment with myocardial infarction (Page 7, 2nd paragraph). 

Comment: "ECV value in patients with AS varied from less than 20% to larger than 70%. Such high ECV indicates that ECV in the myocardium is almost the same as ECV of the blood pool. ECV of 70-80% should be artifactual. ECV of 60-70% is probably due to artifact or presence of infarction".

Response: We thank the reviewer for this comment. We agree that there was a great variation in ECV values among the AS group. However, the great majority of AS patients (80% of them) had an ECV between 26% and 57% (see results section, page 12, 1st paragraph). After re-examining our data we found only 3 patients with an ECV between 61 and 70%, and only 1 patient with an ECV >70%. Therefore, those measurements of a very high ECV were not common. For these patients, systolic pulmonary arterial pressure (SPAP) was very high (average 60.3 mmHg for 3 patients, 1 patient did not have SPAP due to lack of TR). We assume that these particular patients had severe diastolic dysfunction, possibly in the setting of extent myocardial fibrosis, which might have resulted in the high ECV values. Although our ECV measurements were made very cautiously, we cannot completely rule out erroneous measurement in those patients. To note, none of these patients had a history of MI, and all had preserved ejection fraction.

Comment: "Presence of infarction that shows high ECV may be the main source of higher rate of adverse events".

Response: Please see above our response regarding MI in our cohort.

Comment: "Wide range of ECV values in normal subject in this study raises suspicion in its accuracy. ECV values in normal subjects (about 26%) has been well established in both CT and MRI as seen in ref#9".

Response: We thank the reviewer for this comment. The mean ECV of the control group in our study was indeed slightly lower compared with the reference above, however all patients but one had an ECV > 15%. Possible explanation for small differences in ECV value between normal subjects in our cohort and the cohort in the cited reference could be related to the age (median age 57 in our cohort compared with 65 in paper by Kurita et al).

Minor points

Comment: "Line 33; “newly-defined” is misleading because it gives an impression that the authors suggested a new classification in this manuscript"

Response: We thank the reviewer for this comment. The text "newly defined" was changed to "recently published" (see abstract section, line 34).

Comment: "Line 78 - This sentence is not true anymore".

Response: We cited the new publications suggested by the reviewer, please see line 77-79.

Comment: "Line 81; Long-term clinical outcome? With only 12 months of follow up?"

Response: We thank the reviewer for this comment. The phrase long-term was erased (Introduction section, page 5, 3rd paragraph, Line 82).

Comment: "Line 101; Scan protocol and reconstruction of pre-contrast and delayed-phase CT should be described in more details. Conversely, CT protocol for coronary CTA can be shortened for this manuscript".

Response: The protocol of the pre- and post-contrast CT scan was described more in details and the contrast-enhanced scan was shortened. The changes were done in page 6, under CT data acquisition and reconstruction.

Comment: "Line 103; Acquisition at 75% RR can suffer from severe motion artifact in case of high heart rate. Please report heart rate during acquisition".

Response: Mean heart rate for the study cohort was 72.1 beats per minute (±10.9). This sentence was added to the methods section (Materials and methods, CT data acquisition and reconstruction, page 6, line 118).

Comment: "Line 109; Is 50-60ml sufficient for evaluation of delayed enhancement and ECV?"

Response: Our study focused on ECV measurements. The average weight of the study population was 73 kg, which mean that the average contrast agent dose was about 0.8 ml/kg, which is sufficient for ECV calculation mainly in elderly patients with relatively limited GFR. For example, in the study of Abadia et al. (JCCT 2020; 14: 162-167) they used a 70 ml contrast agent despite a higher BMI of their population. 

Comment: "Line 126; Representing the patient’s ECV with a ROI drawn on septum only is probably not adequate".

Response: Please see our response to your 3rd major revision comment above.

Comment: "Line 207; Please clarify what the box and whiskers represent".

Response: In order to avoid any confusion, we deleted the ECV mean value in the figure legend, since the middle line in the box plot represent the median value. 

Comment: "Line 296; This is not the first study"

Response: We thank the reviewer for this comment. At the time of our initial analysis, we were not aware of such studies. Indeed, in the last few months some studies regarding this issue were published. The text "this is the first study" was changed to "our study is one of very few studies" (Discussion, page 16, 1st paragraph).

Comment: "Line 301; “newly described” is misleading".

Response: We thank the reviewer for this comment. The phrase "newly described" was changed to "recently described" (Discussion section, page 16, 1st paragraph) 

Reviewer 2:

Thank you for your helpful comments and suggestions. Below please find the responses that were incorporated into the manuscript.

Comment: "The aortic interventions were performed in 57 of 75 patients. The prognostic value of CT ECV should be demonstrated by showing Kaplan Meier event-free survival curves in 57 patients who had aortic valve interventions".

Response: Our results were demonstrated as a ROC curve to show the impact of ECV value on the outcomes of the patients independently from valve intervention: TAVI, AVR, and balloon valvuloplasty. The ECV calculation was performed in patients with severe AS undergoing CT before intervention, but not all of them underwent intervention. From a clinical point of view it is crucial to demonstrate our results in this way in order to help the clinician to know which patients would benefit from intervention and which patients would not. 

Comment: The staging of AS was successfully performed by echocardiography in only 66 of 75 patients. How were the severity of AS and the indication of aortic valve intervention determined in the patients who enrolled the study?

Response: In our center, severe aortic stenosis is defined as per contemporary valvular heart disease guidelines (symptoms typical of aortic stenosis, mean aortic gradient > 40 mmhg, valve area < 1.0 cm2). All patients were examined at our outpatient clinic and then referred to a dedicated consultation 'heart team' forum that includes a multidisciplinary team of clinical cardiologists, imaging specialists, interventional cardiologists, cardiac surgeons, and geriatricians as required. All patients were assessed by trans-thoracic echocardiography and ECG-gated cardiac CT. Information about this process was added to the manuscript (materials and methods, study cohort, page 6. 2nd paragraph). 

• Baumgartner H, Falk V, Bax JJ, De Bonis M, Hamm C, Holm PJ, et al; ESC Scientific Document Group. 2017 ESC/EACTS Guidelines for the management of valvular heart disease. Eur Heart J. 2017;38(36):2739-91. doi: 10.1093/eurheartj/ehx391

Comment: "Superiority of CT over CMR for the assessment of myocardial ECV fraction, particularly higher spatial resolution, was too much emphasized in the second paragraph of the introduction. In the current study, the ROI was placed in the interventricular septum, indicating that high spatial resolution and SNR improvement by iterative reconstruction are not relevant. Bean gardening artifacts and motion artifacts are major concerns for CT quantification of myocardial ECV fraction".

Response: We thank the reviewer for the comment. We fully agree with the reviewer. We shortened the paragraph and concentrate on the advantages of CT. Iterative reconstruction used by CT and T1 mapping in MRI are not really important for our manuscript and therefore were skipped from the introduction and Ref. 6 and 10 were deleted (Introduction section, page 4, 2nd paragraph). The effect of Beam-hardening artifacts was mentioned in the methods section (page 5, Study cohort subsection) and limitations section (page 17, 2nd paragraph) and this effect was the reason for excluding patients with pacemaker, ICD, AVR or any other metallic objects.

Comment: "The authors emphasized that the implication of ECV fraction using CT has never been studied in patients with severe AS. Please revise the introduction and discussion by citing the following publications".

Response: We thank the reviewer for this comment. At the time of writing this manuscript these studies have not been published yet. These references were incorporated into the manuscript:

- Oda S. Quantification of Myocardial Extracellular Volume With Planning Computed Tomography for Transcatheter Aortic Valve Replacement to Identify Occult Cardiac Amyloidosis in Patients With Severe Aortic Stenosis. Cir Cardiocasc Imaging 2020;13:e010358

- Tamarappoo B, et al. Prognostic Value of Computed Tomography-Derived Extracellular Volume in TAVR Patients With Low-Flow Low-Gradient Aortic Stenosis. JACC Cardiovascular Imaging 2020 Oct 28

- Scully PR, et al. Identifying Cardiac Amyloid in Aortic Stenosis: ECV Quantification by CT in TAVR Patients. JACC Cardiovascular Imaging 2020 Oct 13.

Minor comments

Comment: Abstract. Similar sentences are repeated for the result and conclusion in the abstract. The presentation of the abstract should be more organized

Response: We performed some changes in the abstract to make it more clear and organized.

Comment: "Introduction. Second paragraph. “Another MRI technique used to quantify DIF is T1-mapping, yet, its key limitation is limited spatial resolution”. The authors put too much emphasis on spatial resolution. At a lower concentration of contrast media, contrast discrimination by X-ray CT is not as good as T1 mapping CMR".

Response: We removed the text about T1 mapping from the introduction, based on the comment from the same reviewer (please see our previous response). 

Comment: "Method Study cohort. Please clarify that 10 of 85 AS patients were evaluated due to inadequate CT image quality. Was a prospective study, or a retrospective study that analyzed As patients who had sufficient CT image quality?"

Response: We thank the reviewer for this comment. We re-phrased the "study cohort" section, and we believe it is clearer now (Methods and materials section, study cohort subsection, page 5, 2nd paragraph).

Comment: "CT data acquisition and reconstruction. “Besides the pre-contrast scan, a unique post-contrast scan with the same scan parameters was added”. Delete unique. Please add a brief explanation as to why post-contrast images were acquired at 7 minutes".

Response: We thank the reviewer for this comment. As requested, the word unique was substituted with "additional". The accumulation of contrast agent in the interstitial tissue needs about 7 minutes as has been used by many previous publications (For example Abadia AF et al. J. Cardiovasc comput tomogr. 2020; 14: 162-167). 

Comment: "Result. Table 1. There was a significant difference in age between AS patients and control subjects (57.2±5.6 years in control subjects vs. 80.6±6.8 years in AS patients). Previous studies demonstrated that ECV was significantly influenced by age. In addition, prevalence of CAD was significantly higher in the AS group (p=0.003). The differences in age and the degree of diffuse atherosclerosis substantially influenced the ECV.

Response: We thank the reviewer for this comment. Indeed, the AS group patients were older and had more comorbidities. There is no doubt that some of the difference in ECV between the groups might be attributed to this age difference. However, our data suggests that the AS itself is a major cause of myocardial fibrosis too. We also performed a multivariate analysis using linear regression, which showed that ECV fraction was significantly higher in patients with AS compared to the control group, independent from age, gender, BMI, diabetes mellitus, and hypertension (this appears in the results section, page 13, 1st paragraph).

We add a table that describes this multivariate analysis. Because our paper has many figures already, it was not added to the paper. Should the reviewer wants us to add this table to the manuscript, we will be happy to do so.

Multivariate analysis for comorbidities between the AS group and the control group

Adjusted factor Estimate Std. Error P value

Aortic stenosis 22.19 4.91 <0.0001

Age 0.27 0.17 0.1

Gender 1.61 2.12 0.4

BMI 0.12 0.22 0.5

DM -3.53 2.36 0.1

HTN 0.62 2.72 0.8

BMI: body mass index; DM: diabetes mellitus; HTN; hypertension

Comment: "In this paragraph, the authors also stated that multivariate analysis showed that ECV fraction was significantly higher in patients with AS compared to the control group, independent from age, gender, BMI, diabetes mellitus, and hypertension. However, the number of patients was limited to adequately confirm the independency of these parameters.

Response: Although the number of the included patients were limited, the results were still significant according to our analysis. Please see the table attached for the previous comment.

Comment: "Results. Figure 6B. NYHA FC at 12 months is a categorial parameter and a scatter plot is not suited for Figure 6B".

Response: We fully agree with the reviewer, but some patients have a NYHA FC between 2 and 3 for example, therefore we thought that a scatter plot would be more appropriate. 

Comment: "Results. Figure 7. Comparison between the highest decile of ECV and lowest ductile of ECV seems to be arbitral".

Response: We thank the reviewer for this comment. This figure represents the upper and lower ends of ECV. The message we want to convey, is that it is possible that Gal-3 levels might be associated with ECV fraction. We believe we could not demonstrate a clear relationship between ECV and Gal-3 in our entire cohort because of our low number of patients. This difference which exists only in the upper and lower deciles clearly does not mean that the relationship exists, it is merely suggesting it, as we stated in the results

Comment: "Page 12. “There were no deaths.” Delete all-cause mortality from the combined endpoints to avoid confusion by the readers".

Response: We thank the reviewer for this comment. We agree that since there were no deaths in our study, the outcome of all-cause mortality seems irrelevant. Therefore, we deleted it from our combined outcome.

Comment: "Results. Figure 8. ROC curve. Add cut-off value for ECV".

Response: We thank the reviewer for this comment. Optimal cut-off value. Sensitivity/specificity decision plots yielded an ECV fraction of 40.8% as the optimal cutoff points for prediction of clinical outcomes, with a sensitivity of 91% and specificity of 64%. Patients with an ECV fraction >47.8% had the highest likelihood for the combined clinical outcomes of stroke and hospitalization for heart failure at 12-months, compared to patients with an ECV fraction < 30.5%, (31.6% vs. 10.1%, p=0.02). This text appears in the results section, page 13, 2nd paragraph.

Comment: "Page 14. Discussion. “Patients in the highest deciles of ECV fraction had a significantly higher serum Gal-3 level than patients in the lowest deciles. The combination of two different methods for assessing and quantifying cardiac damage (ECV and Gal-3) enforced our findings”. Comparison between the highest decile of and lowest ductile was quite subjective and this statement had no sufficient rationale".

Response: We thank the reviewer for this comment. We erased the term "enforces" and rephrased that this finding "suggests that ECV fraction could be an important factor…" (Discussion section, page 17, 1st paragraph).

Reviewer 3:

Thank you for your helpful comments and suggestions. Below please find the responses that were incorporated into the manuscript.

Comment: "CTecv was calculated in the septum pre and post contrast as perviously described in CMR and CT manuscripts. As it is supposed to reflect diffuse myocardial fibrosis, the basal septum should be representative as it is relatively spared from myocardial infarction. With that in mind, I am very concerned about the distribution of the CTecv with 11 patients having an ECV >50% and a majority an ECV >40%. This amount of interstitial expansion is rather unusual unless there is evidence of myocardial infiltation (e.g. cardiac amyloidosis). The alternative is a methodological error".

Response: Only 6 patients in the entire group had previous MI and care was taken to avoid ECV measurement in segments with previous MI. In fact, the mean ECV in 6 patients with previous MI was only 37%. Since the basal septum is close to the membranous septum, we decided to use the mid septum for ECV calculation, as performed in previous studies (For example Bandula et al. 2013; 269). The increased ECV value might be related to the AS severity and could be in part the explanation of the high ECV value in our study population. But we have to mention, that the coexistence of cardiac amyloidosis in our study population cannot be fully excluded, this was stated in the study limitation (page 16, line 342-344).

Comment: "The abstract should include structured headings, and the authors should include a heading for the discussion, which is missing". 

Response: We thank the reviewer for this comment. Those mistakes were corrected.

Comment: "The authors have chosen Galactin-3 rather than established cardiac biomarkers like BNP or troponin in this paper. Although this is interesting, the lack of BNP and troponin feels like an oversight".

Response: We thank the reviewer for this comment. Indeed, serum BNP and troponin are important biomarkers in cardiac patients. When designing this trial, we though the most novel and interesting marker to be tested is galectin-3, which was shown to reflect fibrotic processes across the human body. In this trial we did not measure BNP and troponin, and we do not have their levels for our patients, since they are not routinely taken before TAVI/AVR. 

Comment: "The combined clinical endpoint of all-cause mortality, stroke and hospitalization for heart failure is misleading and not ideal: -- although all cause mortality and HHF are adequate endpoints, the link between diffuse fibrosis and stroke is not clear. -- no deaths occured, only HHF and stroke -- the highest quartile of ECV chosen here was ECV>47.8% which is unusually high, and can only be explained by amyloidosis (see above)".

Response: We thank the reviewer for this comment. It is indeed not clear whether this is an incidental finding or not. We added the sentence "The demonstrated relation between stroke and diffuse myocardial fibrosis is not clear at this moment, however it might be related to the fact that of the 3 patients who had a stroke, 2 had atrial fibrillation. The existence of atrial fibrillation in those patients might suggest a possible higher degree of myocardial fibrosis compared to patients without atrial fibrillation" to the discussion section (page 16, last paragraph). As we mentioned previously, we cannot exclude the coexistence of cardiac amyloidosis, however, the increased value of ECV could also be related to the severity of AS and accordingly to the outcomes in this patients. 

Comment: "Regarding endpoints, it is unclear how many endpoints occured periprocedural". 

Response: We thank the reviewer for this comment. In our cohort there were no periprocedural deaths. Among 3 patients who suffered a stroke – one had the event 6 months post procedure, and 2 within the first 6 months after the procedure. None of the events was peri-procedural.

Specific Comments.

Comment: "The authors need to make it clear in the abstract that on 57/75 patients underwent aortic valve intervention (and what intervention TAVR vs SAVR vs valvuloplasty)".

Response: the phrase "Out of the 75 patients in the AS group, 49 underwent TAVI, six surgical AVR, two balloon valvuloplasty, and 18 did not undergo any type of intervention". Was added to the abstract.

Comment: "Introduction: The CMR section does not mention ECV quantification (just T1 mapping, which is used for ECV calculation) - this is mentioned in the first paragraph without mentioning the associated imaging modalities (CMR and CT)". 

Response: Based on the comments of reviewer 2, we skipped T1 mapping from the introduction, since it is not important for the focus of our study. We fully agree with the reviewer that pre-contrast and post-contrast T1 mapping are in parallel with the method used in our study.

Comment: The technological improvements in CT are for certain, though relevant to ECV quantification is not temporal resolution, but Hounsfield unit stability as well as signal to noise ratio. 

Response: We thank the reviewer for the comment and fully agree with it, therefore the importance of temporal resolution was skipped from the introduction, as has also been mentioned by reviewer 2.

Comment: When discussing CMR predictors of outcome in AS, please add ECV by CMR (Everett et al JACC 2020), which is probably more relevant than the LGE papers. 

Response – As requested, this reference was added.

Methods: 

Comment: Please specify ethics committee reference and the period of recruitment.

Response: The study was approved by the ethics committee as we mention that in page 6, line 3rd paragraph. We added the recruitment period "between 2016 and 2018" in page 5, study cohort subsection.

Comment: please specify which contrast agent you administered.

Response: nonionic contrast agent (Iopromide 370; Bayer Schering, Berlin, Germany) was used, we added it in page 7, 1st paragraph.

Comment: please provide more details for the control cohort. Did this cohort undergo CT for a clinical indication or purely for research? This cohort is neither age nor sex matched, and have significantly less hypertension, CAD and renal impairment than the AS cohort. 

Response: The control cohort included patients who were admitted to the hospital for chest pain, and were referred for CT angiography for the purpose of ruling out coronary disease. Before patients were sent to the CT scan, they had to give their consent for an additional post contrast scan that was performed immediately after the routine scan was done. All patients signed an informed consent. This explanation was added to the methods section (Materials and methods, study cohort subsection, page 5).

Indeed, the control cohort is not matched to the study group, obviously because AS patients are very old and tend to have co-morbidities. The purpose of creating this control group was to present the difference in ECV between AS patients and healthy individuals. There is no doubt that some of the difference in ECV between the groups might be attributed to this age difference and possibly to other comorbidities. However, our data suggests that the AS itself is a major cause of myocardial fibrosis too. We also performed a multivariate analysis using linear regression, which showed that ECV fraction was significantly higher in patients with AS compared to the control group, independent from age, gender, BMI, diabetes mellitus, and hypertension (this appears in the results section, page 13, 1st paragraph).

We add a table that describes this multivariate analysis. Because our paper has many figures already, it was not added to the paper. Should the reviewer wants us to add this table to the manuscript, we will be happy to do so.

Multivariate analysis for comorbidities between the AS group and the control group

Adjusted factor Estimate Std. Error P value

Aortic stenosis 22.19 4.91 <0.0001

Age 0.27 0.17 0.1

Gender 1.61 2.12 0.4

BMI 0.12 0.22 0.5

DM -3.53 2.36 0.1

HTN 0.62 2.72 0.8

BMI: body mass index; DM: diabetes mellitus; HTN; hypertension

Results:

Comment: "Please restructure this and place the figure legends to the end of the manuscript.

Response: According to the journal instructions, these legends should be incorporated in the text ("Figure captions must be inserted in the text of the manuscript, immediately following the paragraph in which the figure is first cited (read order). Do not include captions as part of the figure files themselves or submit them in a separate document). We would be happy to change this upon request.

Comment: how many patients had pacemaker or ICDs? These are more likely to affect myocardial HU then AoV Calcification.

Response: While initially recruited 85 patients with symptomatic severe AS, 10 patients were excluded due to pacemakers, implantable cardioverter defibrillators, metallic foreign objects in the proximity of the heart, and surgical aortic valve replacement. All of these may lead to beam-hardening artifacts, which may degrade the inaccuracy in the ECV measurements. This sentence now appears in the manuscript (Materials and methods section, study cohort subsection, page 4)

Discussion:

Comment: please add a heading and restructure the discussion to follow a logical flow.

Response: The first paragraph was divided in 2, the first one summarizes the results of our study and the second one disscuss the importance of ECV evaluation. In addition, we added a separate paragraph for limitation.

Comment: Please add reference and discuss papers on CTecv by Tamarapoo and Scully that address CTecv in TAVI and outcome (both JACC Imaging). 

Response: As requested, these papers were added and discussed..

---

## [Decision Letter · Decision Letter 1]

17 Feb 2021

PONE-D-20-34373R1

Myocardial extracellular volume quantification by computed tomography predicts outcomes in patients with severe aortic stenosis

PLOS ONE

Dear Dr. Hamdan,

Thank you for submitting your manuscript to PLOS ONE. After careful consideration, we feel that it has merit but does not fully meet PLOS ONE’s publication criteria as it currently stands. Therefore, we invite you to submit a revised version of the manuscript that addresses the points raised during the review process.

Please see the additional editor comments below that need to be addressed.

We look forward to receiving your revised manuscript.

Kind regards,

Ify Mordi

Academic Editor

PLOS ONE

Additional Editor Comments (if provided):

A scatterplot is not appropriate for figure 6b - please change - better would be a boxplot as NYHA is definitely categorical.

The lack of troponin and/or BNP needs to be at least mentioned in the limitations - given their use it should be at least discussed as galectin is measured.

The variability in ECV should also be mentioned. The authors should at least note that they do have some high ECV values. They mention that they cannot exclude amyloidosis in the limitations, and they should add that in the context of the identified high ECV values this could be relevant.

The authors should also add in the limitations that the cohort was not matched for age or other important comorbidities such as hypertension.

The final limitation to add is that ECV was only taken from one site rather than a global measure or average, so it is possible that it may not be completely representative of the whole myocardium (accepting that this is the method used in most studies).

Reviewers' comments:

Reviewer's Responses to Questions

**Comments to the Author**

1. If the authors have adequately addressed your comments raised in a previous round of review and you feel that this manuscript is now acceptable for publication, you may indicate that here to bypass the “Comments to the Author” section, enter your conflict of interest statement in the “Confidential to Editor” section, and submit your "Accept" recommendation.

Reviewer #1: All comments have been addressed

2. Is the manuscript technically sound, and do the data support the conclusions?

Reviewer #1: Yes

3. Has the statistical analysis been performed appropriately and rigorously? 

Reviewer #1: Yes

4. Have the authors made all data underlying the findings in their manuscript fully available?

Reviewer #1: Yes

5. Is the manuscript presented in an intelligible fashion and written in standard English?

Reviewer #1: Yes

6. Review Comments to the Author

Reviewer #1: (No Response)

7. PLOS authors have the option to publish the peer review history of their article (what does this mean?). If published, this will include your full peer review and any attached files.

Reviewer #1: No

---

## [Author Response · Author response to Decision Letter 1]

19 Feb 2021

Reviewer 1:

Thank you for your helpful comments and suggestions. Below please find the responses that were incorporated into the manuscript.

Comment: " A scatterplot is not appropriate for figure 6b - please change - better would be a boxplot as NYHA is definitely categorical."

Response: As requested, The correlation between ECV fraction and NYHA class at 12 months is now represented via a boxplot graph. Correlation was measured via a spearman correlation test. See figure 6b

Comment: " The lack of troponin and/or BNP needs to be at least mentioned in the limitations - given their use it should be at least discussed as galectin is measured."

Response: As requested, this issue is now mentioned in the limitation paragraph (page 16, limitation paragraph).

Comment: " The variability in ECV should also be mentioned. The authors should at least note that they do have some high ECV values. They mention that they cannot exclude amyloidosis in the limitations, and they should add that in the context of the identified high ECV values this could be relevant."

Response: As requested, this issue is now mentioned in the limitation paragraph (page 16, limitation paragraph).

Comment: " The authors should also add in the limitations that the cohort was not matched for age or other important comorbidities such as hypertension"

Response: As requested, this issue is now mentioned in the limitation paragraph (page 16, limitation paragraph).

Comment: " The final limitation to add is that ECV was only taken from one site rather than a global measure or average, so it is possible that it may not be completely representative of the whole myocardium (accepting that this is the method used in most studies)".

Response: As requested, this issue is now mentioned in the limitation paragraph (page 16, limitation paragraph).

---

## [Editor Report · Decision Letter 2]

24 Feb 2021

Myocardial extracellular volume quantification by computed tomography predicts outcomes in patients with severe aortic stenosis

PONE-D-20-34373R2

Dear Dr. Hamdan,

We’re pleased to inform you that your manuscript has been judged scientifically suitable for publication and will be formally accepted for publication once it meets all outstanding technical requirements.

Kind regards,

Ify Mordi

Academic Editor

PLOS ONE
---

## [Editor Report · Acceptance letter]

1 Mar 2021

PONE-D-20-34373R2 

Myocardial extracellular volume quantification by computed tomography predicts outcomes in patients with severe aortic stenosis 

Dear Dr. Hamdan:

I'm pleased to inform you that your manuscript has been deemed suitable for publication in PLOS ONE. Congratulations! Your manuscript is now with our production department. 

Kind regards, 

on behalf of

Dr. Ify Mordi 

Academic Editor

PLOS ONE